# IMPROVED OPERATOR LEARNING BY ORTHOGONAL ATTENTION

## ABSTRACT

Neural operators, as an efficient surrogate model for learning the solutions of PDEs, have received extensive attention in the field of scientific machine learning. Among them, attention-based neural operators have become one of the main-streams in related research. However, existing approaches overfit the limited training data due to the considerable number of parameters in the attention mechanism.To address this, we develop an orthogonal attention based on the eigende-composition of the kernel integral operator and the neural approximation of eigen-functions. The orthogonalization naturally poses a proper regularization effect on the resulting neural operator, which aids in resisting overfitting and boosting generalization. Experiments on six standard neural operator benchmark datasets comprising both regular and irregular geometries show that our method can out-perform competing baselines with decent margins.

## 1 INTRODUCTION

Partial Differential Equations (PDEs) are essential tools for modeling and describing intricate dynamics in scientific and engineering domains (Zachmanoglou & Thoe, 1986). Solving the PDEs routinely rely on well-established numerical approaches such as finite element methods (FEM) (Zienkiewicz et al., 2005), finite difference methods (FDM) (Thomas, 2013), spectral methods (Ciarlet, 2002; Courant et al., 1967), etc. Due to the infinite-dimensional nature of the function space, traditional numerical solvers often rely on discretizing the data domain. However, this introduces a balance between efficiency and accuracy: finer discretization offers higher precision but at the expense of greater computational complexity.

Deep learning methods have shown promise in lifting such a trade-off (Li et al., 2020) thanks to their high inference speed and expressiveness. Specifically, physics-informed neural networks (PINNs) (Raissi et al., 2019) first combine neural networks (NNs) with physical principles for PDE solving. Yet, PINNs approximate the solution associated with a certain PDE instance and hence cannot readily adapt to problems with different yet similar setups. By learning a map between the input condition and the PDE solution in a data-driven manner, neural operators manage to solve a family of PDEs, with the DeepONet (Lu et al., 2019) as a representative example. Fourier Neural Operators (FNOs) (Li et al., 2020; Tran et al., 2023; Li et al., 2022; Wen et al., 2022; Grady et al., 2022; Gupta et al., 2021; Xiong et al., 2023) shift the learning to Fourier space to enhance speed while maintaining efficacy through the utilization of the Fast Fourier Transform (FFT). Since the development of attention mechanism (Vaswani et al., 2017), considerable effort has been devoted to developing attention-based neural operators to improve expressiveness and address irregular mesh (Cao, 2021; Li et al., 2023a; Ovadia et al., 2023; Fonseca et al., 2023; Hao et al., 2023; Li et al., 2023b).

Despite the considerable progress made in neural operators, there remain non-trivial challenges in its practical applications. On the one hand, the training targets of neural operators are usually acquired from classical PDE solvers, which can be computationally demanding. For instance, simulations for tasks like airfoils can require about 1 CPU-hour per sample (Li et al., 2022). On the other hand, powerful models are prone to overfitting when trained on small data. A potential remediation is to develop appropriate regularization mechanisms for neural operators to combat overfitting. This can arguably enhance the model's generalization performance, enabling it to generalize to unseen validation data and even across different resolutions and timesteps for time-dependent PDEs.

Given these motivations, we aim to develop a neural operator framework that inherently accommodates proper regularization. We start from the observations that the kernel integral operator involved in Green's function for linear PDEs can be rewritten with orthonormal eigenfunctions. Such an expansion substantially resembles the attention mechanism without softmax while incorporating the orthogonal regularization. Empowered by this, we follow the notion of neural eigenfunctions (Deng et al., 2022b;a) to implement an orthogonal attention module and stack it repeatedly to construct orthogonal neural operator (ONO). As shown in Figure 1, ONO is structured with two disentangled pathways. The bottom one approximates the eigenfunctions through expressive NNs, while the top one specifies the evolvement of the PDE solution. In practice, the orthogonalization operation can be implemented by cheap manipulation of the exponential moving average (EMA) of the feature covariance matrix. It is empirically proven that ONO can generalize substantially better than competitive baselines across both spatial and temporal axes.

To summarize, our contributions are:

- We introduce the novel orthogonal attention, which is inherently integrated with orthogonal regularization while maintaining linear complexity, and detail the theoretical insights.
- We introduce ONO, a neural operator built upon orthogonal attention. It employs two disentangled pathways for approximating the eigenfunctions and PDE solutions.
- We conduct comprehensive studies on six popular operator learning benchmarks and observe satisfactory results: ONO reduces prediction errors by up to 30% compared to baselines and achieves 80% reduction of test error for zero-shot super-resolution on Darcy.

## 2    RELATED WORK

### 2.1    NEURAL OPERATORS

Neural operators map infinite-dimensional input and solution function spaces, allowing them to handle multiple PDE instances without retraining. Following the advent of DeepONet (Lu et al., 2019), the domain of learning neural operator has recently gained much attention. Specifically, DeepONet employs a branch network and a trunk network to separately encode input functions and location variables, subsequently merging them for output computation. FNO (Li et al., 2020) learns the operator in the spectral domain to conjoin good accuracy and inference speed. F-FNO (Tran et al., 2023) improves FNO by integrating separable spectral layers and residual connections, accompanied by a composite set of training strategies. Geo-FNO (Li et al., 2022) employs a map connecting irregular domains and uniform latent meshes to address arbitrary geometries effectively. Numerous alternative variants have been proposed from various perspectives thus far (Grady et al., 2022; Wen et al., 2022; Xiong et al., 2023). However, FNOs are grid-based, leading to increased computational demands for both training and inference as PDE dimensions expand.

Considering the input sequence as a function evaluation within a specific domain, attention operators can be seen as learnable projection or kernel integral operators. These operators have gained substantial research attention due to their scalability and effectiveness in addressing PDEs on irregular meshes. Kovachki et al. (2021) demonstrates that the standard attention mechanism can be considered as a neural operator layer. Galerkin Transformer (Cao, 2021) proposes two self-attention operators without softmax and provides theoretical interpretations for them. HT-Net (Liu et al., 2022) proposes a hierarchical attention operator to solve multi-scale PDEs. GNOT (Hao et al., 2023) proposes a linear cross-attention block to facilitate the encoding of diverse input types. However, despite their promising potential, attention operators are susceptible to overfitting especially when the available training data are rare.

### 2.2    EFFICIENT ATTENTION MECHANISMS

The Transformer model (Vaswani et al., 2017) has gained popularity in diverse domains, including natural language processing (Chen et al., 2018), computer vision (Parmar et al., 2018), and bioinformatics (Rives et al., 2021). However, the vanilla softmax attention encounters scalability issues due to its quadratic space and time complexity. To tackle this, several methods with reduced complexity have been proposed (Child et al., 2019; Zaheer et al., 2020; Wang et al., 2020; Katharopoulos et al., 2020; Xiong et al., 2021). Concretely, sparse Transformer (Child et al., 2019) reduces complexity

by sparsifying the attention matrix. Linear Transformer (Katharopoulos et al., 2020) achieves complexity by replacing softmax with a kernel function. Nyströmformer (Xiong et al., 2021) employs the Nyström method to approximate standard attention, maintaining linear complexity.

In the context of PDE solving, Cao (2021) proposes the linear Galerkin-type attention mechanism, which can be regarded as a trainable Petrov–Galerkin-type projection. OFormer (Li et al., 2023a) develops a linear cross-attention module for disentangling the output and input domains. Fact-Former (Li et al., 2023b) employs axial computation in the attention operator to reduce computational costs. Compared to them, we not only introduce an attention mechanism without softmax at linear complexity but also include an inherent regularization mechanism.

## 3 ORTHOGONAL NEURAL OPERATOR

In this section, we first provide an overview of the orthogonal neural operator and then elaborate on the orthogonal attention mechanism. We will also discuss its theoretical foundations.

### 3.1 PROBLEM SETUP

Operator learning involves learning the map from the input function $f : D \to \mathbb{R}^{d_f} \in \mathcal{F}$ to the PDE solution $u : D \to \mathbb{R}^{d_u} \in \mathcal{U}$, where $D \subset \mathbb{R}^{d_0}$ is a bounded open set. Let $\mathcal{G} : \mathcal{F} \to \mathcal{U}$ denotes the ground-truth solution operator. Our objective is to train a $\boldsymbol{\theta}$-parameterized neural operator $\mathcal{G}_{\boldsymbol{\theta}}$ to approximate $\mathcal{G}$. The training is driven by a collection of function pairs $\{f_i, u_i\}_{i=1}^N$. NN models routinely cannot accept an infinite-dimensional function as input or output, so we discretize $f_i$ and $u_i$ on mesh $\boldsymbol{X} := \{\boldsymbol{x}_j \in D\}_{1 \leq j \leq M}$, yielding $\boldsymbol{f}_i := \{(\boldsymbol{x}_j, f_i(\boldsymbol{x}_j))\}_{1 \leq j \leq M}$ and $\boldsymbol{u}_i := \{(\boldsymbol{x}_j, u_i(\boldsymbol{x}_j))\}_{1 \leq j \leq M}$. We use $\boldsymbol{f}_{i,j}$ to denote the element in $\boldsymbol{f}_i$ that corresponds to $\boldsymbol{x}_j$.

The data fitting is usually achieved by optimizing the following problem:

$$\min_{\boldsymbol{\theta}} \frac{1}{N} \sum_{i=1}^N \frac{\|\mathcal{G}_{\boldsymbol{\theta}}(\boldsymbol{f}_i) - \boldsymbol{u}_i\|_2}{\|\boldsymbol{u}_i\|_2}, \tag{1}$$

where the regular mean-squared error (MSE) is augmented with a normalizer $\|\boldsymbol{u}_i\|_2$ to account for variations in absolute scale across benchmarks following (Li et al., 2020). We refer to this error as $l_2$ relative error in the subsequent sections.

### 3.2 THE MODEL

**Overview.** Basically, an $L$-stage ONO takes the form of

$$\mathcal{G}_{\boldsymbol{\theta}} := \mathcal{P} \circ \mathcal{K}^{(L)} \circ \sigma \circ \mathcal{K}^{(L-1)} \circ \cdots \circ \sigma \circ \mathcal{K}^{(1)} \circ \mathcal{E}, \tag{2}$$

where $\mathcal{E}$ maps $\boldsymbol{f}_i$ to hidden states $\boldsymbol{h}_i^{(1)} \in \mathbb{R}^{M \times d}$, $\mathcal{P}$ projects the states to solutions, and $\sigma$ denotes the non-linear transformation. $\mathcal{K}^{(l)}$ refer to NN-parameterized kernel integral operators following the prior arts in neural operator (Kovachki et al., 2021), which is motivated by the connection between kernel integral operator and Green's function for solving linear PDEs.

Note that $\mathcal{K}^{(l)}$ accepts hidden states $\boldsymbol{h}_i^{(l)} \in \mathbb{R}^{M \times d}$ as input instead of infinite-dimensional functions as in the traditional kernel integral operator. It should also rely on some parameterized configuration of a kernel. FNO defines linear transformations in the spectral space to account for this (Li et al., 2020), but the involved Fourier bases are fixed and data-independent. Instead, we advocate directly parameterizing the kernel in the original space with the help of neural eigenfunctions (Deng et al., 2022b;a). Specifically, we leverage an additional NN to extract hierarchical features from $\boldsymbol{f}_i$, which, after orthogonalization and normalization, suffice to define $\mathcal{K}^{(l)}$.

We outline the overview of ONO in Figure 1, where the two-flow structure is clearly displayed. We pack the orthonormalization step and eigenfunctions-based kernel integral into a module named *orthogonal attention*, which will be detailed later. The decoupled architecture offers significant flexibility in specifying the NN blocks within the bottom flow.

**Encoder.** The encoder is multi-layer perceptrons (MLPs) that accept $\boldsymbol{f}_i$ as input for dimension lifting. Features at every position $\boldsymbol{x}_j$ are extracted separately.

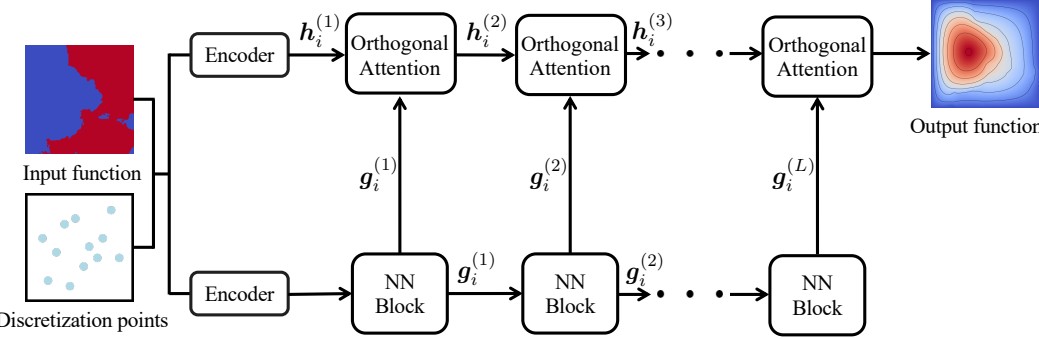

Figure 1: Model overview. There are two flows in ONO. The bottom one extracts expressive features for input data, forming an approximation to the eigenfunctions associated with the kernel integral operators for defining ONO. The top one updates the PDE solutions based on orthogonal attention, which involves linear attention and orthogonal regularization.

**NN Block.** In the bottom flow, the NN blocks are responsible for extracting features, which subsequently specify the kernel integral operators for defining ONO. We can leverage any existing architecture here but focus on transformers due to their great expressiveness. In detail, we formulate the NN block as follow:

$$\tilde{\boldsymbol{g}}_i^{(l)} = \boldsymbol{g}_i^{(l)} + \text{Attn}(\text{LN}(\boldsymbol{g}_i^{(l)})), \quad \boldsymbol{g}_i^{(l+1)} = \tilde{\boldsymbol{g}}_i^{(l)} + \text{FFN}(\text{LN}(\tilde{\boldsymbol{g}}_i^{(l)})), \tag{3}$$

where $\boldsymbol{g}_i^{(l)} \in \mathbb{R}^{M \times d'}$ denotes the output of $l$-th NN block for the data $\boldsymbol{f}_i$. $\text{Attn}(\cdot)$ represents a self-attention module applied over the $M$ positions. $\text{LN}(\cdot)$ indicates layer normalization (Ba et al., 2016). $\text{FFN}(\cdot)$ refers to a two-layer feed forward network. Here, we can freely choose well-studied self-attention mechanisms, e.g., standard attention (Vaswani et al., 2017) and others that enjoy higher efficiency to suit specific requirements (Katharopoulos et al., 2020; Xiong et al., 2021; Cao, 2021).

**Orthogonal Attention.** To prevent overfitting the limited training data, we introduce the orthogonal attention module with orthogonal regularization. As the core of ONO, this module characterizes the evolution of PDE solutions and enhances generalization performance. It transforms the deep features from the NN blocks to orthogonal eigenmaps, based on which the kernel integral operators are constructed and the hidden states of PDE solutions are updated. Concretely, we first project the NN features $\boldsymbol{g}_i^{(l)} \in \mathbb{R}^{M \times d'}$ to:

$$\hat{\boldsymbol{\psi}}_i^{(l)} = \text{ort}(\hat{\boldsymbol{g}}_i^{(l)}) = \text{ort}(\boldsymbol{g}_i^{(l)} \boldsymbol{w}_Q^{(l)}) \in \mathbb{R}^{M \times k}, \tag{4}$$

where $\boldsymbol{w}_Q^{(l)} \in \mathbb{R}^{d' \times k}$ is a trainable weight. $\text{ort}(\cdot)$ is the orthonormalization operation which renders each column of $\hat{\boldsymbol{\psi}}_i^{(l)}$ correspond to the evaluation of a specific neural eigenfunction on $\boldsymbol{f}_i$.

Given these, the orthogonal attention update the hidden states $\boldsymbol{h}_i^{(l)}$ of PDE solutions via:

$$\tilde{\boldsymbol{h}}_i^{(l+1)} = \hat{\boldsymbol{\psi}}_i^{(l)} \text{diag}(\hat{\boldsymbol{\mu}}^{(l)})[\hat{\boldsymbol{\psi}}_i^{(l)\top}(\boldsymbol{h}_i^{(l)} \boldsymbol{w}_V^{(l)})], \tag{5}$$

where $\boldsymbol{w}_V^{(l)} \in \mathbb{R}^{d \times d}$ is a trainable linear weight to refine the hidden states, and $\hat{\boldsymbol{\mu}}^{(l)} \in \mathbb{R}_+^k$ denote positive eigenvalues associated with the induced kernel and are trainable in practice. This update rule is closely related to Mercer's theorem, as will be detailed in Section 3.3.

The non-linear transformation $\sigma$ is implemented following the structure of the tranditional attention mechanism, which involves residual connections (He et al., 2016) and FFN transformation:

$$\boldsymbol{h}_i^{(l+1)} = \text{FFN}(\text{LN}(\tilde{\boldsymbol{h}}_i^{(l+1)} + \boldsymbol{h}_i^{(l)})). \tag{6}$$

The FFN in the final orthogonal attention serves as $\mathcal{P}$ to map hidden states to PDE solutions.

**Implementation of $\text{ort}(\cdot)$.** As mentioned, we leverage $\text{ort}(\cdot)$ to make $\hat{\boldsymbol{g}}_i^{(l)}$ follow the structure of the outputs of eigenfunctions. We highlight that the orthonormalization lies in the function space,

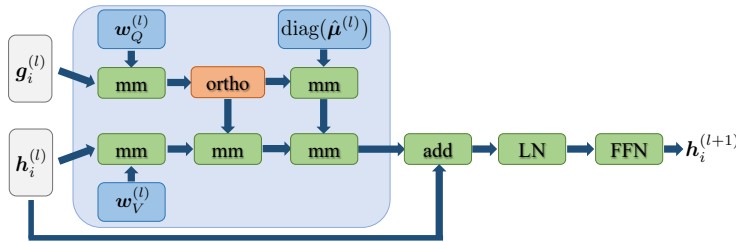

Figure 2: Orthogonal Attention Module Architecture: The module incorporates matrix multiplications ("mm") and an orthogonalization process ("ortho"). The output of the NN block, denoted as $\boldsymbol{g}_i^l$, and the hidden state of the input function, represented as $\boldsymbol{h}_i^l$, undergo processing as shown in Equation 5. Following this, the module includes a residual connection, layer normalization, and a two-layer FFN.

i.e., among the output dimensions of the function $\hat{g}^{(l)} : \boldsymbol{f}_{i,j} \mapsto \hat{\boldsymbol{g}}_{i,j}^{(l)} \in \mathbb{R}^k$ instead of the column vectors. Thereby, we should not orthonormalize matrix $\hat{\boldsymbol{g}}_i^{(l)}$ over its columns but manipulate $\hat{g}^{(l)}$.

To achieve this, we first estimate the covariance over the output dimensions of $\hat{g}^{(l)}$, which can be approximated by Monte Carlo (MC) estimation:

$$\mathbf{C}^{(l)} \approx \frac{1}{NM} \sum_{i=1}^N \sum_{j=1}^M [\hat{g}^{(l)}(\boldsymbol{f}_{i,j})^\top \hat{g}^{(l)}(\boldsymbol{f}_{i,j})] = \frac{1}{NM} \sum_{i=1}^N [\hat{\boldsymbol{g}}_i^{(l)\top} \hat{\boldsymbol{g}}_i^{(l)}]. \tag{7}$$

Then, we orthonormalize $\hat{g}^{(l)}$ by right multiplying the matrix $\mathbf{L}^{(l)-\top}$, where $\mathbf{L}^{(l)}$ is the lower-triangular matrix arising from the Cholesky decomposition of $\mathbf{C}^{(l)}$, i.e., $\mathbf{C}^{(l)} = \mathbf{L}^{(l)} \mathbf{L}^{(l)\top}$.

In the vector formula, there is

$$\hat{\boldsymbol{\psi}}_i^{(l)} := \hat{\boldsymbol{g}}_i^{(l)} \mathbf{L}^{(l)-\top}. \tag{8}$$

The covariance of the functions producing $\hat{\boldsymbol{\psi}}_i^{(l)}$ can be approximately estimated:

$$\frac{1}{NM} \sum_{i=1}^N \left[ \left( \hat{\boldsymbol{g}}_i^{(l)} \mathbf{L}^{(l)-\top} \right)^\top \hat{\boldsymbol{g}}_i^{(l)} \mathbf{L}^{(l)-\top} \right] = \mathbf{L}^{(l)-1} \mathbf{C}^{(l)} \mathbf{L}^{(l)-\top} = \mathbf{I}, \tag{9}$$

which conforms that these functions can be regarded as orthonormal neural eigenfunctions that implicitly define a kernel.

However, in practice, the model parameters evolve repeatedly, we cannot trivially estimate $\mathbf{C}^{(l)}$, which involves the whole training set, at a low cost per training iteration. Instead, we propose to approximately estimate $\mathbf{C}^{(l)}$ via the exponential moving average trick—similar to the update rule in batch normalization (Ioffe & Szegedy, 2015), we maintain a buffer tensor $\mathbf{C}^{(l)}$ and update it with training mini-batches. We reuse the recorded training statistics to ensure the stability of inference.

The above process incurs a cubic complexity w.r.t. $k$ due to the Cholesky decomposition. However, the overall complexity of the proposed orthogonal attention remains linear w.r.t the number of measurement points $M$ as usually $k \ll M$.

## 3.3 THEORETICAL INSIGHTS

This section provides the theoretical insights behind orthogonal attention. We abuse notations when there is no misleading. Consider a kernel integral operator $\mathcal{K}$ taking the form of:

$$(\mathcal{K}h)(\boldsymbol{x}) := \int_D \kappa(\boldsymbol{x}, \boldsymbol{x}') h(\boldsymbol{x}') \, d\boldsymbol{x}', \quad \forall \boldsymbol{x} \in D, \tag{10}$$

where $\kappa$ is a positive semi-definite kernel and $h$ is the input function. Given $\psi_i$ as the eigenfunction of $\mathcal{K}$ corresponding to the $i$-th largest eigenvalue $\mu_i$, we have:

$$\int_D \kappa(\boldsymbol{x}, \boldsymbol{x}') \psi_i(\boldsymbol{x}') \, d\boldsymbol{x}' = \mu_i \psi_i(\boldsymbol{x}), \quad \forall i \geq 1, \forall \boldsymbol{x} \in D$$
$$\langle \psi_i, \psi_j \rangle = \mathbb{1}[i = j], \quad \forall i, j \geq 1, \tag{11}$$

where $\langle a, b \rangle := \int a(\boldsymbol{x}) b(\boldsymbol{x}) \, d\boldsymbol{x}$ denotes the inner product in $D$. By Mercer's theorem, there is:

$$(\mathcal{K}h)(\boldsymbol{x}) = \int_D \kappa(\boldsymbol{x}, \boldsymbol{x}') h(\boldsymbol{x}') \, d\boldsymbol{x}' = \int \sum_{i \geq 1} \mu_i \psi_i(\boldsymbol{x}) \psi_i(\boldsymbol{x}') h(\boldsymbol{x}') \, d\boldsymbol{x}' = \sum_{i \geq 1} \mu_i \langle \psi_i, h \rangle \psi_i(\boldsymbol{x}). \tag{12}$$

Although we cannot trivially estimate the eigenfunctions $\psi_i$ in the absence of $\kappa$'s expression, Equation 12 offers us new insights on how to parameterize a kernel integral operator. In particular, we can truncate the summation in Equation 12 and introduce a parametric model $\hat{\psi}(\cdot) : D \to \mathbb{R}^k$ with orthogonal outputs and build a neural operator $\hat{\mathcal{K}}$ with the following definition:

$$(\hat{\mathcal{K}}h)(\boldsymbol{x}) := \sum_{i=1}^{k} \langle \hat{\psi}_i, h \rangle \hat{\psi}_i(\boldsymbol{x}). \tag{13}$$

We demonstrate the convergence of $\hat{\mathcal{K}}$ towards the ground truth $\mathcal{K}$ under MSE loss in the Appendix A. In practice, we first consider $\boldsymbol{X} := \{\boldsymbol{x}_j\}_{1 \leq j \leq M}$ and $\boldsymbol{Y} := \{\boldsymbol{x}_j\}_{1 \leq j \leq M'}$ as two sets of measurement points to discretize the input and output functions. We denote $\hat{\boldsymbol{\psi}} \in \mathbb{R}^{M \times k}$ and $\hat{\boldsymbol{\psi}}' \in \mathbb{R}^{M' \times k}$ as the evaluation of the model $\hat{\psi}$ on $\boldsymbol{X}$ and $\boldsymbol{Y}$ respectively. Futhermore, let $\boldsymbol{h} \in \mathbb{R}^M$ represent the evaluation of $h$ on $\boldsymbol{X}$. There is:

$$(\hat{\mathcal{K}}h)(\boldsymbol{Y}) \approx \sum_{i=1}^{k} [\hat{\psi}_i(\boldsymbol{X})^\top h(\boldsymbol{X})] \hat{\psi}_i(\boldsymbol{Y}) = \hat{\boldsymbol{\psi}}' \hat{\boldsymbol{\psi}}^\top \boldsymbol{h}. \tag{14}$$

Comparing Equation 12 and Equation 13, we can see that the scaling factors $\mu_i$ are omitted, which may undermine the model flexibility in practice. To address this, we introduce a learnable vector $\hat{\boldsymbol{\mu}} \in \mathbb{R}_+^k$ to Equation 14, resulting in:

$$(\hat{\mathcal{K}}h)(\boldsymbol{Y}) \approx \hat{\boldsymbol{\psi}}' \mathrm{diag}(\hat{\boldsymbol{\mu}}) \hat{\boldsymbol{\psi}}^\top \boldsymbol{h}. \tag{15}$$

As shown, there is an attention structure—$\hat{\boldsymbol{\psi}}' \mathrm{diag}(\hat{\boldsymbol{\mu}}) \hat{\boldsymbol{\psi}}^\top$ corresponds to the attention matrix that defines how the output function evaluations attend to the input. Besides, the orthonormalization regularization arises from the nature of eigenfunctions, benefitting to alleviate overfitting and boost generalization. When $\boldsymbol{X} = \boldsymbol{Y}$, the above attention structure is similar to regular self-attention mechamism with a symmetric attention matrix. Otherwise, it boils down to a cross-attention, which enables our approach to query output functions at arbitrary locations independent of the inputs. Find more details regarding this in Appendix A.

## 4 EXPERIMENTS

We conduct extensive experiments on diverse and challenging benchmarks across various domains to showcase the effectiveness of our method.

**Benchmarks.** We first evaluate our model's performance on Darcy and NS2d (Li et al., 2020) benchmarks to evaluate its capability on regular grids. Subsequently, we extend our experiments to benchmarks with irregular geometries, including Airfoil, Plasticity, and Pipe, which are represented in structured meshes, as well as Elasticity, presented in point clouds (Li et al., 2022).

**Baselines.** We compare our model with several baseline models, including the well-recognized FNO (Li et al., 2020) and its variants Geo-FNO (Li et al., 2022), F-FNO (Tran et al., 2023), and U-FNO (Wen et al., 2022). Furthermore, we consider other models such as Galerkin Transformer (Cao, 2021), LSM (Wu et al., 2023), and GNOT (Hao et al., 2023). It's worth noting that LSM and GNOT are the latest state-of-the-art (SOTA) neural operators.

**Implementation details.** We use the $l_2$ relative error in Equation 1 as the training loss and evaluation metric. We train all models for 500 epochs. Our training process employs the AdamW optimizer (Loshchilov & Hutter, 2018) and the OneCycleLr scheduler (Smith & Topin, 2019). We initialize the learning rate at $10^{-3}$ and explore batch sizes within the range of $\{2, 4, 8, 16\}$. Unless specified otherwise, we choose either the Linear transformer block from (Katharopoulos et al., 2020) or the Nyström transformer block from (Xiong et al., 2021) as the NN block in our model. Our experiments are conducted on a single NVIDIA RTX 3090 GPU.

Table 1: The main results on six benchmarks compared with seven baselines. Lower scores indicate superior performance, and the best results are highlighted in bold. "*" means that the results of the method are reproduced by ourselves. "-" means that the baseline cannot handle this benchmark.

| MODEL | NS2d | Airfoil | Pipe | Darcy | Elasticity | Plasticity |
|---|---|---|---|---|---|---|
| FNO | 0.1556 | - | - | 0.0108 | - | - |
| Galerkin | 0.1401 | - | - | 0.0084 | - | - |
| Geo-FNO | 0.1556 | 0.0138 | 0.0067 | 0.0108 | 0.0229 | 0.0074 |
| U-FNO* | 0.2182 | 0.0137 | 0.0050 | 0.0266 | 0.0226 | **0.0028** |
| F-FNO* | 0.1213 | 0.0079 | 0.0063 | 0.0318 | 0.0316 | 0.0048 |
| LSM* | 0.1693 | 0.0062 | 0.0049 | **0.0069** | 0.0225 | 0.0035 |
| GNOT | 0.1380 | 0.0076 | - | 0.0105 | **0.0086** | - |
| Ours | **0.1195** | **0.0056** | **0.0034** | 0.0072 | 0.0118 | 0.0048 |

Table 2: Zero-shot super-resolution results on Darcy. The model is trained on data of $43 \times 43$ resolution.

| MODEL | $s = 61$ | $s = 85$ | $s = 141$ | $s = 211$ | $s = 421$ |
|---|---|---|---|---|---|
| FNO | 0.1164 | 0.1797 | 0.2679 | 0.3160 | 0.3631 |
| Ours | **0.0204** | **0.0259** | **0.0315** | **0.0349** | **0.0386** |

Table 3: Results on different time intervals on NS2d.

| MODEL | Seen | Unseen |
|---|---|---|
| FNO | 0.0982 | 0.2446 |
| Ours | **0.0889** | **0.2143** |

### 4.1 MAIN RESULTS

Table 1 reports the results. Remarkably, our model achieves SOTA performance on three of these benchmarks, reducing the average prediction error by $13\%$. Specifically, it reduces the error by $31\%$ and $10\%$ on Pipe and Airfoil, respectively. In the case of NS2d, which involves temporal predictions, our model surpasses all baselines. We attribute it to the temporal generalization enabled by our orthogonal attention. We conjecture that the efficacy of orthogonal regularization contributes to our model's excellent performance in these three benchmarks by mitigating overfitting the limited training data. These three benchmarks encompass both regular and irregular geometries, demonstrating the versatility of our model across various geometric settings.

Furthermore, our model achieves the second-lowest prediction error on Darcy and Elasticity benchmarks, albeit with a slight margin compared to the SOTA baselines. We notice that our model and the other attention operator (GNOT) demonstrate a significant reduction in error when compared to other operators that utilize a learnable mapping to convert the irregular input domain into or back from a uniform mesh. This mapping process can potentially introduce errors. However, attention operators naturally handle irregular meshes for sequence input without requiring mapping, leading to superior performance. Our model also exhibits competitive performance on plasticity, involving the mapping of a shape vector to the complex mesh grid with a dimension of deformation. These results highlight the versatility and effectiveness of our model as a framework for learning operators.

### 4.2 GENERALIZATION EXPERIMENTS

We conduct experiments on the generalization performance in both the spatial and temporal axes. First, a zero-shot super-resolution experiment is conducted on Darcy. The model is trained on $43 \times 43$ resolution data and evaluated on resolutions up to nearly ten times that size ($421 \times 421$). Subsequently, we train the model to predict timesteps 11-18 and evaluate it on two subsequent intervals: timesteps 11-18, denoted as "Seen", and timesteps 19-20, denoted as "Unseen". We choose the FNO (Li et al., 2020) as the baseline due to its use of the orthogonal fourier basis functions, which may potentially offer regularization benefits.

The results are shown in Table 2 and Table 3. On Darcy, the prediction error of FNO increases dramatically as the evaluation resolution grows. In contrast, our model exhibits a much slower increase in error and maintains a low prediction error even with excessively enlarged resolution, notably reducing the prediction error by $89\%$ compared to FNO on the $421 \times 421$ resolution. On NS2d, Our model outperforms in both time intervals, reducing the prediction error by $9\%$ and $12\%$. We further visualize some generalization results in these two scenarios in Figure 3 and Figure 4. The results are consistent with the reported values. These results demonstrate that our model exhibits remarkable

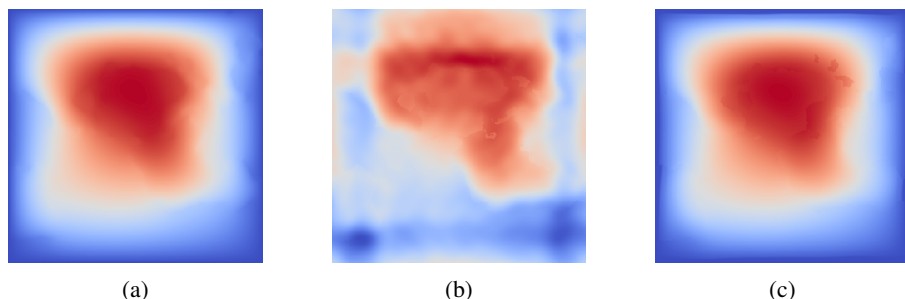

Figure 3: Zero-shot super-resolution results on Darcy. (a): Groundtruth. (b): Prediction of FNO. (c): Prediction of ONO. Trained on $43 \times 43$ data and evaluated on $421 \times 421$.

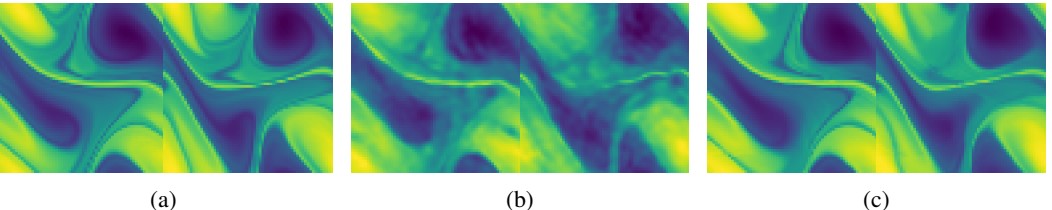

Figure 4: The prediction on timesteps 19 and 20 of models trained to predict timesteps 11-18 on NS2d. (a): Groundtruth. (b): Prediction of FNO. (c): Prediction of ONO.

generalization capabilities in both temporal and spatial domains. Acquiring high-resolution training data can be computationally expensive. Our model's mesh-invariant property enables effective high-resolution performance after being trained on low-resolution data, potentially resulting in significant computational cost savings.

### 4.3 ABLATION EXPERIMENTS

We conduct a detailed ablation study to assess the effectiveness of various components in ONO on Airfoil, Elasticity, and Pipe.

**Necessity of Orthogonalization.** A significant distinguishing characteristic of our orthogonal attention mechanism, as opposed to other attention mechanisms, is the inclusion of the orthogonalization process. To investigate the indispensability of this process, we carry out a series of experiments on three benchmarks. "BN" and "LN" denote the batch normalization (Ioffe & Szegedy, 2015) and the layer normalization (Ba et al., 2016), while "Ortho" signifies the orthogonalization process. It's worth noting that the attention mechanism coupled with layer normalization assumes a structure resembling Fourier-type attentinon (Cao, 2021).

As shown in Table 4, our orthogonal attention consistently outperforms other attention mechanisms across all benchmarks, resulting in a remarkable reduction of prediction error, up to $81\%$ on Airfoil and $39\%$ on Pipe. We conjecture that the orthogonalization may benefit model training through feature scaling. Additionally, the inherent linear independence among orthogonal eigenfunctions aids the model in effectively distinguishing between various features, contributing to its superior performance compared to the conventional normalizations.

**Influence of NN Block.** To show the compatibility of our model, we conduct experiments with different NN blocks. We choose the Galerkin transformer block in operator learning (Cao, 2021) and two linear transformer blocks in other domains, including the Linear transformer block in (Xiong et al., 2021) and the Nyström transformer block in (Katharopoulos et al., 2020).

Table 5 showcases the results. The Nyström transformer block performs better on all three benchmarks and reduces the error up to $43\%$ on Pipe. We notice that the Linear transformer and Galerkin transformer are both kernel-based methods transformer methods. The Nyström attention uses a downsampling approach to approximate the softmax attention, which aids in capturing positional relationships and contributes to the feature extraction. However, all the variants consistently exhibit competitive performance, showcasing the flexibility and robustness of our model.

Table 4: Ablations on the orthogonalization.

| DESIGNS | Airfoil | Elasticity | Pipe |
|---|---|---|---|
| BN | 0.0808 | 0.0149 | 0.2151 |
| LN | 0.0288 | 0.2579 | 0.0056 |
| Ortho | **0.0056** | **0.0118** | **0.0034** |

Table 5: Influence of different NN blocks.

| DESIGNS | Airfoil | Elasticity | Pipe |
|---|---|---|---|
| Linear | 0.0079 | 0.0137 | 0.0060 |
| Nystrom | **0.0056** | **0.0118** | **0.0034** |
| Galerkin | 0.0122 | 0.0133 | 0.0075 |

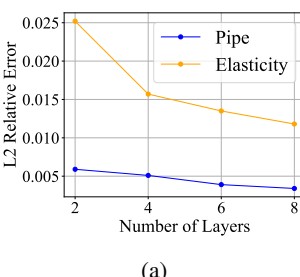 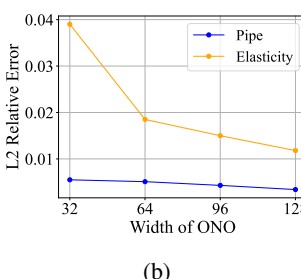 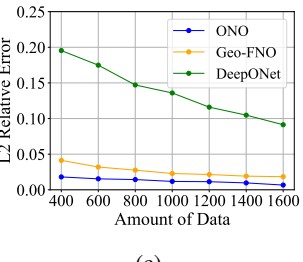

| (a) | (b) | (c) |
|---|---|---|

Figure 5: Results on scaling experiments. (a): Different number of layers. (b): Different widths of the neural operator. (c): Different training data amounts.

## 4.4 SCALING EXPERIMENTS

Our model's architecture offers scalability, allowing adjustments to both its depth and width for enhanced performance or reduced computational costs. We conduct scaling experiments to examine how the prediction error changes with the number of data, the number of layers, and width.

Figure 5a depicts the change in prediction error with an increasing number of layers, while Figure 5b shows how the error responds to a growth in width of ONO. It is evident that error reduction correlates positively with both the number of layers and width. Nevertheless, diminishing returns become apparent when exceeding four layers or a width of 64 on Elasticity. Consequently, we recommend employing a model configuration consisting of four layers and a width of 64 due to its favorable balance between performance and computational cost.

To further assess the scalability of our model, we increase the number of layers to 30 and the learnable parameters to 10 million while keeping the width at 128. We compare it to the 8-layer model, which has approximately 1 million parameters. The results are in Table 6. We denote the models as "ONO-30" and "ONO-8" respectively. The prediction of ONO-30 exhibits a remarkable decrease in both benchmarks, achieving reductions of 37% and 76%.

Table 6: Comparisons between ONO-30 and ONO-8.

| Model | Elasticity | Plasticity |
|---|---|---|
| ONO-8 | 0.0118 | 0.0048 |
| ONO-30 | **0.0047** | **0.0013** |

We assess performance using limited training data in the context of the Elasticity benchmark, as depicted in Figure 5c. We choose Geo-FNO (Li et al., 2022) and DeepONet (Lu et al., 2019), two well-studied neural operators, as our baselines. ONO consistently outperforms the baselines and exhibits a smaller decrease in performance as the amount of training data decreases. Remarkably, even with only 25% of the data used, ONO achieves a similar relative error (0.0181 with 400 training data) compared to Geo-FNO (0.0184 with 1600 training data). These results highlight the inherent regularization capability of ONO, which effectively prevents overfitting on limited data.

## 5 CONCLUSION

This paper aims to addresses the overfitting challenges stemming from the limited training data obtained through classical PDE solvers. Our main contribution is the introduction of regularization mechanisms for neural operators to enhance generalization performance. We propose an attention mechanism with orthogonalization regularization based on the kernel integral rewritten by orthonormal eigenfunctions. We further present a neural operator called ONO, built upon this attention mechanism. Extensive experiments demonstrate the superiority of our approach compared to baselines. This work contributes to mitigating overfitting issues, particularly in the context of large models for solving PDEs, which tend to overfit the limited training data.

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

# A    Theoretical Supplement

**Proof of the convergence of $\hat{\mathcal{K}}$.** To push $\hat{\mathcal{K}}$ in Equation 13 towards to unknown ground truth $\mathcal{K}$, we solve the following minimization problem:

$$
\min_{\hat{\psi}} \ell, \; l := \mathbb{E}_{h \sim p(h)} \left( \int \left[ \sum_{i=1}^{k} \langle \hat{\psi}_i, h \rangle \hat{\psi}_i(\boldsymbol{x}) - (\mathcal{K}h)(\boldsymbol{x}) \right]^2 d\boldsymbol{x} \right)
$$

$$
s.t. : \langle \hat{\psi}_i, \hat{\psi}_j \rangle = \mathbb{1}[i = j], \quad \forall i, j \in [1, k], \tag{16}
$$

We next prove that the above learning objective closely connects to eigenfunction recovery. To show that, we first reformulate the above loss:

$$
\begin{aligned}
\ell &= \mathbb{E}_{h \sim p(h)} \left( \sum_{i=1}^{k} \sum_{i'=1}^{k} \langle \hat{\psi}_i, h \rangle \langle \hat{\psi}_{i'}, h \rangle \langle \hat{\psi}_i, \hat{\psi}_{i'} \rangle - 2 \sum_{i=1}^{k} \langle \hat{\psi}_i, h \rangle \langle \hat{\psi}_i, \mathcal{K}h \rangle + C \right) \\
&= \mathbb{E}_{h \sim p(h)} \left( \sum_{i=1}^{k} \sum_{i'=1}^{k} \langle \hat{\psi}_i, h \rangle \langle \hat{\psi}_{i'}, h \rangle \mathbb{1}[i = i'] - 2 \sum_{i=1}^{k} \langle \hat{\psi}_i, h \rangle \langle \hat{\psi}_i, \mathcal{K}h \rangle + C \right) \\
&= \mathbb{E}_{h \sim p(h)} \left( \sum_{i=1}^{k} \langle \hat{\psi}_i, h \rangle^2 - 2 \sum_{i=1}^{k} \langle \hat{\psi}_i, h \rangle \langle \hat{\psi}_i, \mathcal{K}h \rangle + C \right) \\
&= \mathbb{E}_{h \sim p(h)} \left( \sum_{i=1}^{k} \langle \hat{\psi}_i, h \rangle^2 - 2 \sum_{i=1}^{k} \langle \hat{\psi}_i, h \rangle \left[ \sum_{j \geq 1} \mu_j \langle \psi_j, h \rangle \langle \hat{\psi}_i, \psi_j \rangle \right] + C \right)
\end{aligned} \tag{17}
$$

where $C$ denotes a constant agnostic to $\hat{\psi}$.

Represent $\hat{\psi}_i$ with its coordinates $\boldsymbol{a}_i := [a_{i,1}, \dots, a_{i,j}, \dots]$ in the space spanned by $\{\psi_j\}_{j \geq 1}$, i.e., $\hat{\psi}_i = \sum_{j \geq 1} \boldsymbol{a}_{i,j} \psi_j$. Thereby, $\langle \hat{\psi}_i, \hat{\psi}_{i'} \rangle = \boldsymbol{a}_i^\top \boldsymbol{a}_{i'} := \sum_{j \geq 1} a_{i,j} a_{i',j}$ and $\boldsymbol{a}_i^\top \boldsymbol{a}_{i'} = \mathbb{1}[i = i']$. Likewise, we represent $h$ with coordinates $\boldsymbol{a}_h := [a_{h,1}, \dots, a_{h,j}, \dots]$. Let $\boldsymbol{\mu} := [\mu_1, \mu_2, \dots]$ and $\mathbf{A}_h := \mathbb{E}_{h \sim p(h)} \boldsymbol{a}_h \boldsymbol{a}_h^\top$. There is (we omit the above constant)

$$
\begin{aligned}
\ell &= \mathbb{E}_{h \sim p(h)} \left( \sum_{i=1}^{k} (\boldsymbol{a}_i^\top \boldsymbol{a}_h)^2 - 2 \sum_{i=1}^{k} (\boldsymbol{a}_i^\top \boldsymbol{a}_h) \left[ \sum_{h \geq 1} \mu_j \boldsymbol{a}_{h,j} \boldsymbol{a}_{i,h} \right] \right) \\
&= \mathbb{E}_{h \sim p(h)} \left( \sum_{i=1}^{k} (\boldsymbol{a}_i^\top \boldsymbol{a}_h)^2 - 2 \sum_{i=1}^{k} (\boldsymbol{a}_i^\top \boldsymbol{a}_h)(\boldsymbol{a}_i^\top \mathrm{diag}(\boldsymbol{\mu}) \boldsymbol{a}_h) \right) \\
&= \mathbb{E}_{h \sim p(h)} \left( \sum_{i=1}^{k} \boldsymbol{a}_i^\top (\boldsymbol{a}_h \boldsymbol{a}_h^\top) \boldsymbol{a}_i - 2 \sum_{i=1}^{k} \boldsymbol{a}_i^\top (\boldsymbol{a}_h \boldsymbol{a}_h^\top) \mathrm{diag}(\boldsymbol{\mu}) \boldsymbol{a}_i \right) \\
&= \sum_{i=1}^{k} \boldsymbol{a}_i^\top \left[ \mathbb{E}_{h \sim p(h)} \boldsymbol{a}_h \boldsymbol{a}_h^\top \right] \boldsymbol{a}_i - 2 \sum_{i=1}^{k} \boldsymbol{a}_i^\top \left[ \mathbb{E}_{h \sim p(h)} \boldsymbol{a}_h \boldsymbol{a}_h^\top \right] \mathrm{diag}(\boldsymbol{\mu}) \boldsymbol{a}_i \\
&= \sum_{i=1}^{k} \left[ \boldsymbol{a}_i^\top \mathbf{A}_h \boldsymbol{a}_i - 2 \boldsymbol{a}_i^\top \mathbf{A}_h \mathrm{diag}(\boldsymbol{\mu}) \boldsymbol{a}_i \right] \\
&= \sum_{i=1}^{k} \left[ \boldsymbol{a}_i^\top \mathbf{A}_h \boldsymbol{a}_i - \boldsymbol{a}_i^\top \mathbf{A}_h \mathrm{diag}(\boldsymbol{\mu}) \boldsymbol{a}_i - \boldsymbol{a}_i^\top \mathrm{diag}(\boldsymbol{\mu}) \mathbf{A}_h \boldsymbol{a}_i \right] \\
&= \sum_{i=1}^{k} \boldsymbol{a}_i^\top \left[ \mathbf{A}_h - \mathbf{A}_h \mathrm{diag}(\boldsymbol{\mu}) - \mathrm{diag}(\boldsymbol{\mu}) \mathbf{A}_h \right] \boldsymbol{a}_i.
\end{aligned} \tag{18}
$$

$\mathbf{A}_h$ and $\mathbf{A}_h - \mathbf{A}_h \mathrm{diag}(\boldsymbol{\mu}) - \mathrm{diag}(\boldsymbol{\mu}) \mathbf{A}_h$ are both symmetric positive semidefinite matrices with infinity rows and columns. Considering the orthonoramlity constraint on $\{\boldsymbol{a}_i\}_{i=1}^{k}$, minimizing $\ell$ will push $\{\boldsymbol{a}_i\}_{i=1}^{k}$ towards the $k$ eigenvectors with smallest eigenvalues of $\mathbf{A}_h - \mathbf{A}_h \mathrm{diag}(\boldsymbol{\mu}) -$

$\mathrm{diag}(\boldsymbol{\mu})\mathbf{A}_h$. In the case that $\mathbf{A}_h$ equals to the identity matrix, i.e., $\mathbb{E}_{h\sim p(h)}\langle h,\psi_i\rangle\langle h,\psi_j\rangle = \mathbb{1}[i=j]$, there is :

$$\ell = \sum_{i=1}^{k} \boldsymbol{a}_i^{\top}\left[\mathbf{A}_h - \mathbf{A}_h\mathrm{diag}(\boldsymbol{\mu}) - \mathrm{diag}(\boldsymbol{\mu})\mathbf{A}_h\right]\boldsymbol{a}_i = k - 2\sum_{i=1}^{k}\boldsymbol{a}_i^{\top}\mathrm{diag}(\boldsymbol{\mu})\boldsymbol{a}_i. \tag{19}$$

Then $\{\boldsymbol{a}_i\}_{i=1}^{k}$ will converge to the $k$ principal eigenvectors of $\mathrm{diag}(\boldsymbol{\mu})$, i.e., the one-hot vectors with $i$-th element equaling 1. Given that $\hat{\psi}_i = \sum_{j\geq 1}\boldsymbol{a}_{i,j}\psi_j$, the deployed parametric model $\hat{\psi}$ will converge to the $k$ principal eigenfunctions of the unknown ground-truth kernel integral operator.

**Cross-attention Variant.** For a pair of functions $(\boldsymbol{f}_n, \boldsymbol{u}_n)$, the data points used to discretize them are different, denoted as $\mathbf{X}$ and $\mathbf{Y}$. Let $\mathcal{H}^{(l)}, l \in [1, L]$ denote the specified operators at various modeling stages. We define the propagation rule as

$$
\begin{aligned}
(\mathcal{H}^{(1)}\boldsymbol{h}^{(1)})(\mathbf{Y}) &\approx \mathrm{FFN}(\mathrm{LN}(\left(\hat{\boldsymbol{\psi}}^{(1)}(\mathbf{Y})\hat{\boldsymbol{\psi}}^{(1)}(\mathbf{X})^{\top}\left[\boldsymbol{h}^{(1)}(\mathbf{X})\right]\right))) \\
(\mathcal{H}^{(l)}\boldsymbol{h}^{(l)})(\mathbf{Y}) &\approx \mathrm{FFN}(\mathrm{LN}(\left(\hat{\boldsymbol{\psi}}^{(l)}(\mathbf{Y})\hat{\boldsymbol{\psi}}^{(l)}(\mathbf{Y})^{\top}\left[\boldsymbol{h}^{(l)}(\mathbf{Y})\right] + \boldsymbol{h}^{(l)}(\mathbf{Y})\right))), \; l \in [2, L]
\end{aligned}
\tag{20}
$$

where $\mathrm{FFN}(\cdot)$ denotes a two-layer FFN and $\mathrm{LN}(\cdot)$ denotes the layer normalization.

## B  HYPERPARAMETERS AND DETAILS FOR MODELS

**FNO and its Variants.**

For FNO and its variants (Geo-FNO, F-FNO, U-FNO), we employ 4 layers with modes of 12 and widths from $\{20, 32\}$. Notably, Geo-FNO reverts to the vanilla FNO when applied to benchmarks with regular grids, resulting in equivalent performance for Darcy and NS2d benchmarks. For U-FNO, the U-Net path is appended in the last layer. FNO-2D is implemented in generalization experiments. The batch size is selected from $\{10, 20\}$.

**LSM.** We configure the model with 8 basis operators and 4 latent tokens. The width of the first scale is set to 32, and the downsampling ratio is 0.5. The batch size is selected from $\{10, 20\}$.

Table 7: Comparison of Run Time per epoch on the Darcy benchmark. "Galerkin" denotes the Galerkin Transformer, "Fourier" denotes the Fourier Transformer, "LA" denotes the Linear transformer block, and "NA" denotes the Nyström transformer block.

| MODEL | FNO | Galerkin | Fourier | LSM | ONO (LT) | ONO (NT) |
|---|---|---|---|---|---|---|
| Run Time(s) | 3.81 | 9.88 | 58.16 | 9.05 | 7.87 | 34.46 |

**Run Time Comparison.**

Table 7 compares the run times of different neural operators. All models are trained with a batch size of 8 on Darcy. FNO, LSM, and ONO are fixed as 4 layers. The width of ONO is set to 128, and the number of eigenfunctions $k$ is 16. According to the results, ONO with the Linear transformer block achieves the second-highest speed and completes each epoch with a 13% reduction in time compared to LSM. ONO with the Nyström transformer block exhibits a lower speed, surpassing only the Fourier Transformer, which has a square computational complexity. Notably, the time cost primarily stems from the NN block in the bottom pathway rather than the orthogonal attention module in the top pathway.

