# OpenReview forum: "Improved Operator Learning by Orthogonal Attention"
_ICLR.cc/2024/Conference — Submitted to ICLR 2024_

### Official Review · Reviewer_TYEx · 2023-11-03

**Soundness:** 3 good
**Presentation:** 4 excellent
**Contribution:** 3 good
**Rating:** 6
**Confidence:** 3

**Summary:**

The paper proposes an attention based operator learning method, where the features are orthonormalized at each layer. The orthonormalized features imply a trainable kernel and also act as a built-in regularization mechanism that aims to improve the generalization and prevent overfitting.

**Strengths:**

The paper is overall well-written and focuses on the important problem of generalization and overfitting in operator learning. The theoretical build-up is insightful and the experiments show promising results, although some details are lacking.

**Weaknesses:**

- The model complexity is not completely clear. As pointed out by the authors, the Cholesky decomposition and also inverting the matrix $L$ are computationally expensive. While this complexity depends on the dimension $k$ of $\hat{g}_i$, the choice of $k$ is not clear in the experiments. (I assume that the *width* in the experiments refers to the size of $d$ in $h_i^l$s.)
- While the paper tries to address the overfitting with limited data, the experimental results concerning the dataset size are limited to only one dataset and one baseline method. A similar comparison with other methods, and if possible, with DeepONet, would help understanding the data efficiency and generalization power of these methods better.

**Questions:**

1. I would like to see more details and clarifications regarding the choices made for hyperparameters in the experiments, such as the size of $k$ and $d'$.
2. A per-epoch runtime comparison with other methods would also help clarify the model complexity.
3. I am wondering whether multi-head attention can be naturally applied in the Orthogonal Attention blocks and if the authors have considered or tried that.
4. Right before Equation 6, the non-linearity is said to come after the residual connections and FFN. However, Equation 6 doesn't explicitly apply $\sigma$. Does that mean that $\sigma$ is built into the last layer of the FFN? If so, since the FFN is said to serve as the projection $\mathcal{P}$ to the solutions in the last layer, are the last layer's outputs also passed to $\sigma$? Wouldn't that be troublesome for the predictions?
5. Figure 2 is a bit confusing and misleading when compared with Equation 5. A more detailed caption might help clarify the process.

---

> ### Author Response · Authors · 2023-11-15
> **Response to Reviewer TYEx**
>
> Thank you for your detailed, helpful feedback! We're glad that you found our paper to be well-written and insightful in terms of the theoretical foundation. We address your concerns point by point below.
>
> Q1 : Details and clarifications about the choices of hyperparameters.
>
> The size of $d'$ is always fixed to be equal to $d$ in $h_{i}^{(l)}$ to reduce the choice of hyperparamters in our experiments. $d'$ is typically 8 (for Elasticity, Plasticity, Pipe and Airfoil) or 16 (for Darcy and NS2d), which is much less than the width of the neural operator (128).
>
> Q2: Runtime comparison.
>
> In the revision, we provide the runtime comparison on the Darcy benchmark in the **appendix B**. The results are shown as below:
>
> | Model | FNO | Galerkin | Fourier | LSM |ONO (LT)| ONO (NT) |
> |:---------|:---------:|:---------:|:---------:|:---------:|:---------:|:---------:|
> | Run Time (s) | 3.81 |  9.88 | 58.16 | 9.05  | 7.87 | 34.46 |
>
> Our implementation includes two types of NN blocks: Linear Transformer ("LT") and Nystrom Transformer ("NT"). The results demonstrate that the time cost of our neural operator is acceptable compared to the baselines, and  the complexity primarily originates from the NN block. We believe that these findings provide valuable insights into the model complexity. Thanks to your suggestion.
>
> Q3: Applying multi-head attention.
>
> We think it's an insightful suggestion to improve our proposed attention. We assume that the orthogonalization process is applied after the input is transformed into multiple attention heads. Theoretically, it will become kernel integral operator with multiple kernels, which may enrich the ability. We leave this as an interesting direction for future work.
>
> Q4: The position of the activation function.
>
> We noticed that the sentence can be misleading and revised in the revision. We clarify that the activation function is applied after the first linear layer of the two-layer FFN and a simple linear layer without activation function is applied in the last layer.
>
> Q5: The caption of Figure 2.
>
> We notice that Equation 5 can be misleading in the computation order. In practice, we will compute as $\tilde{h}^{(l+1)}_{i}= \hat{\psi}^{(l)}_i\mathrm{diag}(\hat{\boldsymbol{\mu}}^{(l)})[{\hat{\psi}^{(l)}_i}{}^{\top}( h^{(l)}_i w^{(l)}_V)]$ to maintain a linear complexity in our attention module. We have revised both the caption and the equation in the revision. Thanks to your comment!
>
> Thanks to your useful feedback again! Given these encouraging new results, we hope you might be convinced to increase your score. We appreciate your feedback strengthening our paper!

---

### Official Review · Reviewer_bpG7 · 2023-11-03

**Soundness:** 3 good
**Presentation:** 2 fair
**Contribution:** 2 fair
**Rating:** 5
**Confidence:** 2

**Summary:**

The paper has proposed a new attention method to improve the generalization ability of neural operators for PDE.

**Strengths:**

1. The method is easy to understand with some theory backup.
2. The method has shown improved performance across many datasets.

**Weaknesses:**

1. It is unclear how to justify that the method mitigates the overfitting problem, which is one of the central claims of the paper.
2. The impact of the method versus the models' size should be studied.

**Questions:**

1. what are the performance versus model sizes?
2. how do you measure the overfitting?

**Details Of Ethics Concerns:**

No concern.

---

> ### Author Response · Authors · 2023-11-15
> **Response to Reviewer bpG7**
>
> Thank you for your useful suggestions! We appreciate your positive feedback on the clarity and comprehensibility of our method. We address your concerns point by point below.
>
> Q1: Performance versus model sizes.
>
> We have conducted experiments about the width and the number of layers of our model in our paper. Please see **section 4.4** for more details. The results show that our neural operator improves as we increase the width or layers. We further increased the layers to 30 and parameters to 10M, leading to notable enhancements on both the Elasticity and Plasticity benchmarks.
>
> Q2: The measurement of the overfitting.
>
> We mainly demonstrate the issue of overfitting by comparing the performance of our proposed neural operator with varying amounts of training data against the baseline Geo-FNO. In the latest revision of our paper, we have incorporated another well-acknowledged neural operator, DeepONet, as an additional baseline for comparison. The results highlight that our model achieves comparable performance while requiring significantly less training data. Additionally, our model exhibits lower performance degradation compared to the baselines. We also experimented with increasing the number of layers to 30 on the Elasticity and Plasticity benchmark. Our neural operator demonstrates significant improvement without exhibiting overfitting. This is in contrast to the performance of Geo-FNO, which shows a decline in performance when the number of layers exceeded 4 in [1]. This observation suggests that our proposed neural operator offers an effective regularization mechanism that contributes to its enhanced performance compared to the baseline.
> We believe that these results effectively showcase the problem of overfitting and highlight the regularization benefits of our proposed orthogonal attention.
>
> We sincerely hope for a reconsideration of our paper's score. If you have any more questions or find some mistakes, please feel free to correct us.
>
> [1] Alasdair Tran, Alexander Mathews, Lexing Xie, and Cheng Soon Ong. Factorized fourier neural
> operators. In The Eleventh International Conference on Learning Representations, 2023

---

> > ### Author Response · Authors · 2023-11-21
> > **Sincerely looking forward to the further discussions**
> >
> > Dear reviewer,
> >
> > We are wondering if our response and revision have resolved your concerns. If our response has addressed your concerns, we would highly appreciate it if you could re-evaluate our work and consider raising the score.
> >
> > If you have any additional questions or suggestions, we would be happy to have further discussions.
> >
> > Best regards,
> >
> > The Authors

---

> ### Author Response · Authors · 2023-11-23
> **Thanks for reviewing our paper**
>
> Dear reviewer bpG7,
>
> As the discussion session draws to a close, we would like to inquire if there are any additional comments or clarifications you would like to make. We are more than willing to provide responses to any inquiries and address any feedback you may have. Thanks!

---

### Official Review · Reviewer_rVud · 2023-11-04

**Soundness:** 2 fair
**Presentation:** 2 fair
**Contribution:** 2 fair
**Rating:** 6
**Confidence:** 3

**Summary:**

This paper proposes a novel neural operator architecture called Orthogonal Neural Operator (ONO) for solving partial differential equations (PDEs).
The key idea is to introduce an orthogonal attention mechanism that provides inherent regularization to combat overfitting when training on limited PDE data from classical numerical solvers.
The orthogonal attention is motivated by representing the kernel integral operator for PDE solutions using orthonormal eigenfunctions based on Mercer's theorem.
This eigendecomposition perspective allows defining a parameterized attention-like module with orthogonal regularization on the features.
Specifically, ONO consists of two pathways - one pathway uses neural networks to extract expressive features that approximate eigenfunctions, while the other pathway updates the PDE solution states based on the orthogonal attention.
The orthogonal attention first projects the eigenfunction-like features to an orthonormal space and then performs linear attention weighted by eigenvalues to update the solution states.
The orthogonalization acts as regularization and is implemented efficiently using the covariance matrix and its Cholesky decomposition.
The experiments are conducted on 6 benchmark datasets with regular and irregular geometries.
The empirical evaluations validate the effectiveness of the proposed technique over competitive baselines.

**Strengths:**

- This work proposes a novel orthogonal attention mechanism for neural operators that provides inherent regularization. The connection to eigendecomposition of the kernel operator is an original perspective. The authors introduces a two-pathway architecture with eigenfunction approximation and orthogonal attention-based solution update. The disentangled design is innovative.
- Orthogonal regularization through orthogonalization of features is an interesting way to mitigate overfitting in neural operators.
- The results demonstrate ONO achieves competitive performance by reducing the prediction error substantially compared to prior neural operator methods like FNO, Geo-FNO, LSM, etc. Further analysis shows ONO generalizes remarkably better than baselines for zero-shot super-resolution and predicting unseen time intervals. Ablation studies verify that the orthogonal attention is crucial to the performance gains.
- Broad applicability to diverse PDEs with different geometries highlights the potential of the idea.

**Weaknesses:**

- The motivation of avoiding overfitting with regularization is reasonable, but the paper lacks experiments that directly demonstrate overfitting issues in baseline models to substantiate the need for orthogonal regularization. Adding such empirical analysis could strengthen the motivation.
- While the eigendecomposition perspective provides insights, the connection to eigenfunctions is mainly conceptual. More theoretical analysis that formally relates the orthogonal attention to spectral properties could enhance the rigor.
- The ablation study verifies the usefulness of the orthogonalization component. However, it does not isolate the impact of the two-pathway architecture itself. Additional experiments are needed to demonstrate the benefits of the disentangled design.
- The long-term impact could be boosted by testing on real-world datasets and problems beyond standard benchmarks to showcase effectiveness in complex practical settings. The paper could also provide a more comprehensive evaluation of the model's performance across a broader spectrum of PDEs, including those with more complex boundary conditions and non-linearities. This would not only demonstrate the robustness of the model but also identify potential limitations in its current form.

**Questions:**

Please see above.

---

> ### Author Response · Authors · 2023-11-15
> **Response to Reviewer rVud**
>
> Thank you for your helpful suggestions! We are glad that you think the orthogonal attention is "novel" and the two-pathway architecture is innovative. We address your concerns point by point below.
>
> Q1: Empirical analysis of overfitting.
>
> Thanks to your comments about this. We have performed an experiment on the Elasticity benchmark, varying the number of training data, to illustrate the issue of overfitting when data is limited. Our results indicate that our neural operator exhibits a smaller decrease in performance with fewer training data compared to the baseline model Geo-FNO. We have also included another well-acknowledged neural operator, DeepONet, as an additional baseline in our revised version. This addition serves to further highlight the issue and strengthen the motivation behind our work. What's more, we have also experimented with increasing the number of layers to 30 on the Elasticity benchmark. In contrast, Geo-FNO showed a decline in performance when exceeding 4 layers in [1]. This suggests that our proposed neural operator incorporates effective regularization mechanisms, contributing to its superior performance compared to the baseline model.
>
> Q2: Theoretical analysis of orthogonal attention.
>
> Our proposed attention is based on the insight of eigendecomposition. We have also included theoretical support in the form of Eq (17) and Eq (18) in the **appendix A**, which demonstrates that the parametric $\hat{\psi}$ will converge to the top-$k$ principal eigenfunctions of the unknown ground-truth kernel integral operator. We think it's interesting and inspiring to relate our attention to spectral properties and really appreciate your suggestion.
>
> Q3: The impact of the disentangled design.
>
> Thanks for your interest in the disentangled design. We think that the orthogonal attention naturally requires two pathways and their inputs: the eigenfunctions and the input function, making it challenging to replace the disentangled design.
>
> Q4: Comprehensive evaluation in complex settings.
>
> We agree that more complex practical settings can evaluate the neural operator more comprehensively. We are also interested in the performance in practical scenarios. However, we emphasize that our current empirical studies align with the related works in this field and can prove the effectiveness of our method. We will explore more complex tasks in the next version.
>
> We welcome any questions or corrections you may have and sincerely hope for a reconsideration of our paper's score. Your feedback is highly valuable to us.
>
> [1] Alasdair Tran, Alexander Mathews, Lexing Xie, and Cheng Soon Ong. Factorized fourier neural
> operators. In The Eleventh International Conference on Learning Representations, 2023

---

### Official Review · Reviewer_DnDW · 2023-11-05

**Soundness:** 3 good
**Presentation:** 3 good
**Contribution:** 3 good
**Rating:** 6
**Confidence:** 3

**Summary:**

the paper proposed a novel network architecture for neural operators that solve PDEs. The proposed architecture incorporates an orthogonal attention mechanism to alleviate the potential overfitting issues, and improve the generalization of the operator for solving PDEs. The demonstration of the effectiveness in the experimental section is clear, and the ablation study highlights the positive contribution of the proposed orthogonal attention mechanism.

**Strengths:**

1. a novel attention mechanism is proposed to address the issue of overfitting, and the experimental sections show improvement over the existing attention mechanisms.

2. the ablation study compares the proposed orthogonal attention mechanism to other normalization schemes, and shows the advantages of the proposed mechanism.

3. scaling up the neural network with the proposed orthogonal attention mechanism brings in performance improvement, which shows that the proposed mechanism improves generalization of neural networks with various sizes.

**Weaknesses:**

1. I understand the motivation that the top line in figure 1 updates the PDE solution so that strong regularization is needed, and that is why the proposed orthogonal attention is incorporated, along with linear attention. However, technically, both the top line and the bottom line are simply nonlinear functions, so have the authors tried to incorporate the proposed attention into both lines to see if it further improves the generalization?

2. the current parametrization requires solving a Cholesky decomposition for each batch of the data during training, and I wonder whether there could be ways to simplify the process. For example, could we fix the orthonormal matrix before training, and learn mu as a function of the input data? in that way, we could avoid taking the inverse of a whole matrix each iteration but rather taking the inverse of individual values in mu.

**Questions:**

n/a

---

> ### Author Response · Authors · 2023-11-15
> **Response to Reviewer DnDW**
>
> Thank you for the valuable feedback! We greatly appreciate your recognition of the novelty of our attention mechanism. We would like to address your concerns point by point as follows.
>
> Q1: Incorporating the orthogonal attention into both lines.
>
> Thanks to this comment. We think that the bottom line requires expressive structures to extract features from the input data and approximate the eigenfunctions rather than structures with regularization that may limit the overall expressiveness of the neural operator. In fact, we conducted such an experiment on the Darcy benchmark and observed that the neural operator exhibits unsatisfactory performance (0.036 versus 0.0072 on Darcy). Hence, we recommend employing a well-studied linear attention instead of orthogonal attention in the bottom line.
>
> Q2: Simplifying the orthogonalization process.
>
> We appreciate your insightful advice regarding simplifying the orthogonalization process. We understand your concern about the complexity involved in the process. The complexity of our proposed process mainly relies on the Cholesky decomposition of the covariance matrix and matrix inversion. We think that both processes maintain an acceptable complexity of O($8^3$) (for Elasticity, Plasticity, Pipe, and Airfoil) and O($16^3$) (for Darcy and NS2d). In our latest revision, we have provided the time cost compared with baselines in **appendix B**, which demonstrates that the complexity is reasonable. We acknowledge that the process can be further simplified and we are grateful for your suggestion. We will explore this in our future work.
>
> If you have any more questions or find some mistakes, please feel free to correct us. We sincerely hope that you will reconsider and potentially increase the score for our paper.

---

### Author Response · Authors · 2023-11-21
**Comment to all reviewers**

We appreciate that reviewers think our work is "novel" (Reviewer DnDW), "an original perspective" (Reviewer eVud), "easy to understand"(Reviewer bpG7), and "insightful" (Reviewer TYEx). We would like to express our gratitude for the valuable feedback, which has significantly strengthened our paper (with the revision now posted).

In response to concerns raised by multiple reviewers regarding overfitting issues in baseline models, we incorporated another well-acknowledged baseline, DeepONet, in experiments conducted with varying amounts of training data in **section 4.4**. Our model shows decreased performance degradation as training data is reduced compared to the baselines. The results also highlight that our model achieves comparable performance while requiring significantly less training data (e.g., 0.0181 with 400 training data compared to Geo-FNO's 0.0184 with 1600 training data). These outcomes substantiate that our method enhances data efficiency and generalization.

We noted that reviewers have also raised concerns regarding the complexity associated with orthogonalization. We presented a runtime comparison with four baselines on the Darcy benchmark in **appendix B**, revealing that the computational time of our model is lower than that of another linear attention-based neural operator, the Galerkin Transformer. The results suggest that the time cost of our neural operator is acceptable, and the complexity primarily originates from the NN block rather than orthogonalization.

We hope that with these changes, we address all major concerns.

---

### Meta-Review · Area_Chair_MSZP · 2023-12-15

**Metareview:**

The paper addresses the challenge of overfitting in attention-based neural operators used for learning solutions of Partial Differential Equations (PDEs). Existing approaches face overfitting issues due to the abundance of parameters in attention mechanisms. In response, the authors propose an orthogonal attention approach grounded in the eigendecomposition of the kernel integral operator and neural approximation of eigenfunctions. The orthogonalization introduces a natural regularization effect on the resulting neural operator, effectively mitigating overfitting and enhancing generalization. Experimental results on six standard neural operator benchmark datasets, encompassing both regular and irregular geometries, demonstrate the superior performance of the proposed method compared to competing baselines.

The paper's rationale for introducing orthogonal regularization to mitigate overfitting is sound, yet empirical evidence demonstrating overfitting issues in baseline models is lacking, diminishing the strength of the motivation. The conceptual link between the eigendecomposition perspective and eigenfunctions is acknowledged, but the paper could benefit from a more thorough theoretical analysis establishing a formal connection between orthogonal attention and spectral properties.

While the ablation study successfully confirms the utility of the orthogonalization component, it falls short of isolating the impact of the two-pathway architecture. Additional experiments are warranted to elucidate the benefits derived from the disentangled design.

To bolster its long-term impact, the paper should extend its evaluation to real-world datasets and problems beyond standard benchmarks, showcasing the model's effectiveness in practical applications. Furthermore, a more extensive assessment across various PDEs, including those featuring complex boundary conditions and non-linearities, would not only highlight the robustness of the model but also uncover potential limitations in its current form.

The reviewers had mixed opinions about the paper, tending a bit more toward acceptance. However, due to the competitive nature of the conference and my own reading of the reviews and the paper (the reviewers with a score of 6 also had several points of valid criticism), I tend a bit more toward a rejection.

**Justification For Why Not Higher Score:**

Several weaknesses where pointed out, even by the reviewers with more positive reviews.

**Justification For Why Not Lower Score:**

-

---

### Decision · Program_Chairs · 2024-01-16

Reject